# Self-Healing Thiolated Pillar[5]arene Films Containing Moxifloxacin Suppress the Development of Bacterial Biofilms

**DOI:** 10.3390/nano12091604

**Published:** 2022-05-09

**Authors:** Dmitriy N. Shurpik, Yulia I. Aleksandrova, Olga A. Mostovaya, Viktoriya A. Nazmutdinova, Regina E. Tazieva, Fadis F. Murzakhanov, Marat R. Gafurov, Pavel V. Zelenikhin, Evgenia V. Subakaeva, Evgenia A. Sokolova, Alexander V. Gerasimov, Vadim V. Gorodov, Daut R. Islamov, Peter J. Cragg, Ivan I. Stoikov

**Affiliations:** 1A.M.Butlerov Chemical Institute, Kazan Federal University, Kremlevskaya, 18, 420008 Kazan, Russia; a.julia.1996@mail.ru (Y.I.A.); olga.mostovaya@mail.ru (O.A.M.); n-vika-art@mail.ru (V.A.N.); reginatzv@gmail.com (R.E.T.); alexander.gerasimov@kpfu.ru (A.V.G.); 2Institute of Physics, Kazan Federal University, Kremlevskaya, 18, 420008 Kazan, Russia; murzakhanov.fadis@yandex.ru (F.F.M.); marat.gafurov@kpfu.ru (M.R.G.); 3Institute of Fundamental Medicine and Biology, Kazan Federal University, Kremlevskaya, 18, 420008 Kazan, Russia; pasha_mic@mail.ru (P.V.Z.); zs_zs97@mail.ru (E.V.S.); zhenya_mic@mail.ru (E.A.S.); 4Enikolopov Institute of Synthetic Polymeric Materials, Russian Academy of Sciences (ISPM RAS), Profsoyuznaya, 70, 117393 Moscow, Russia; gorodovvv@ispm.ru; 5Laboratory for Structural Analysis of Biomacromolecules, Kazan Scientific Center of Russian Academy of Sciences, Lobachevskogo, 2/31, 420111 Kazan, Russia; daut1989@mail.ru; 6School of Applied Sciences, University of Brighton, Huxley Building, Brighton BN2 4GJ, UK; p.j.cragg@brighton.ac.uk

**Keywords:** pillar[5]arene, self-healing, moxifloxacin hydrochloride, electron spin resonance, polymer films, polythiols, antibacterial activity

## Abstract

Polymer self-healing films containing fragments of pillar[5]arene were obtained for the first time using thiol/disulfide redox cross-linking. These films were characterized by thermogravimetric analysis and differential scanning calorimetry, FTIR spectroscopy, and electron microscopy. The films demonstrated the ability to self-heal through the action of atmospheric oxygen. Using UV–vis, 2D 1H-1H NOESY, and DOSY NMR spectroscopy, the pillar[5]arene was shown to form complexes with the antimicrobial drug moxifloxacin in a 2:1 composition (logK11 = 2.14 and logK12 = 6.20). Films containing moxifloxacin effectively reduced *Staphylococcus aureus* and *Klebsiella pneumoniae* biofilms formation on adhesive surfaces.

## 1. Introduction

In recent decades, methods of disrupting pathogenic biofilms have been actively developed [1]. These methods are based on finding ways to inhibit and control the bacterial biofilms formation [2]. It is believed that up to 80% of all human bacterial infections (cystic fibrosis pneumonia, otitis media, pathology of teeth and periodontal tissues, osteomyelitis, urinary tract infections, etc.) are associated with the establishment of biofilms by pathogenic and opportunistic microorganisms [3,4]. Along with this, the formation of microbial biofilms speeds up the corrosion of metals, makes medical equipment unusable, and leads to a deterioration in sanitation and hygiene in medical institutions [5].

To date, three main strategies have been proposed in the struggle against pathogenic biofilms: prevention of bacterial adhesion to the surface [6,7]; disruption of biofilm development and/or impact on its structure with an antimicrobial drug; and impact on the creation of the biofilm, followed by its degradation [8,9,10]. However, all these methods have common limitations, namely a low efficiency on account of the rapid growth and formation of the extracellular bacterial matrix and a short duration of action due to the presence in biofilms of metabolically inactive cells that are insensitive to factors, primarily antimicrobial drugs [11]. The use of polymeric compositions capable of long-term inhibition of biofilm formation will make it possible to eliminate these limitations [12]. Over the past few years, special attention has been given to supramolecular polymer systems formed by macrocyclic structures [13,14]. Such polymer systems can have a number of the required functions, including self-regeneration, controlled adhesion of microbial cells, and the formation of host–guest systems (macrocycle/antibiotic or antiseptic), to inhibit the activity of cells inside the biofilm.

Derivatives of cyclodextrins [15], cucurbit[n]urils [16], and calix[n]arenes [17,18] are frequently used to form supramolecular polymer ensembles with desired properties. Although these macrocycles have a relatively low toxicity, they are quite difficult to modify into polyfunctional polymer structures.

The use of paracyclophane derivatives—pillar[n]arenes—as a macrocyclic system solves this problem [19]. Pillar[n]arenes can be easily functionalized thorough free hydroxyl groups [20,21], and the presence of a hydrophobic cavity promotes the formation of host–guest systems [22,23,24,25]. Recent studies [26,27,28,29,30] demonstrate the effectiveness of using pillar[5]arenes as platforms to create self-assembling drug delivery systems, stimulus-responsive polymeric systems, and antibacterial coatings.

In this work, we developed an original strategy for creating disulfide-containing self-healing materials based on the copolymerization of pillar[5]arene-tetrakis(3-mercaptopropionate) capable of forming host–guest complex with therapeutic drugs that inhibit the development of Gram-positive and Gram-negative bacteria, *Staphylococcus aureus,* and *Klebsiella pneumoniae*.

## 2. Materials and Methods

### 2.1. Characterization

^1^H NMR, ^13^C NMR, and ^1^H-^1^H NOESY spectra were obtained on a Bruker Avance-400 spectrometer (^13^C{^1^H}—100 MHz and ^1^H—400 MHz). Chemical shifts were determined against the signals of residual protons of deuterated solvent (CDCl_3_).

Attenuated total internal reflectance IR spectra were recorded with Spectrum 400 (Perkin Elmer) Fourier spectrometer. The IR spectra from 4000 to 400 cm^−1^ were considered in this analysis. The spectra were measured with 1 cm^−1^ resolution and 64 scans co-addition.

Elemental analysis was performed with Perkin Elmer 2400 Series II instrument.

Mass spectra (MALDI-TOF) were recorded on an Ultraflex III mass spectrometer in a 4- nitroaniline matrix. Melting points were determined using a Boetius Block apparatus.

Additional control of the purity of compounds and monitoring of the reaction were carried out by thin-layer chromatography using Silica G, 200 µm plates, UV 254.

Stationary electron paramagnetic resonance (EPR) spectra were obtained at a frequency of 9.6 GHz (X-band) on a Bruker Elexsys E580 spectrometer at room temperature (modulation amplitude M = 0.1 Gs, microwave power P = 2 μW). Low-temperature experiments were carried out on a Bruker ESP-300 spectrometer using a flow-through helium cryostat.

The pulsed EPR spectra were recorded by the method of detecting the EPR spectrum from the integrated intensity of the electron spin echo (ESE) as a function of the external magnetic field B_0_. Khan’s sequence was used:π/2–τ–π–τ–ESE(1)
where the duration of a π/2 pulse was t_p_ = 16 ns, π pulse t_p_ = 32 ns, and delay between pulses τ = 200 ns.

A two-pulse Hahn sequence was used to determine the transverse relaxation time T_2_. The time interval between pulses τ was increased with a step of 4 ns to the required value, and each time, the integrated ESE intensity was recorded at B = B_0_. Next, we plotted the dependence of the integrated ESE intensity as a function of time 2τ and approximated it with the function:I(2τ) = I_0_ exp(2τ/T_2_)(2)

The transverse relaxation time T_M_ (phase coherence time) was determined.

Irradiation took place for 1 h on an X-ray unit URS-55 (tungsten anti-cathode W, voltage U = 50 kV). The estimated absorption dose is approximately equal to 10 kGy.

^1^H diffusion-ordered spectroscopy (DOSY) spectra were recorded on a Bruker Avance 400 spectrometer at 9.4 tesla at a resonating frequency of 400.17 MHz for ^1^H using a BBO Bruker 5 mm gradient probe. The temperature was regulated at 298 K, and no spinning was applied to the NMR tube. DOSY experiments were performed using the STE bipolar gradient pulse pair (stebpgp1s) pulse sequence with 16 scans of the 16 data points collected. The maximum gradient strength produced in the z direction was 5.35 Gmm^−1^. The duration of the magnetic field pulse gradients (δ) was optimized for each diffusion time (Δ) in order to obtain a 2% residual signal with the maximum gradient strength. The values of δ and Δ were 1.800 μs and 100 ms, respectively. The pulse gradients were incremented from 2 to 95% of the maximum gradient strength in a linear ramp.

### 2.2. Fluorescence Spectroscopy

Fluorescence spectra were recorded on a Fluorolog 3 luminescent spectrometer (Horiba Jobin Yvon). The excitation wavelength was selected as 335 nm. The emission scan range was 350–550 nm. Excitation and emission slits were 2 nm for solutions and 2 nm for supramolecular films. Quartz cuvettes with optical path length of 10 mm were used. Fluorescence spectra were automatically corrected by the Fluoressence program. The spectra were recorded in the solvent system (THF:CH_3_OH = 100:1) with concentration of Moxifloxacin Hydrocloride (moxi) 5 μM. The obtained molar ratio of polymers to moxi **3/3S** or **3/4S** was 1:10. The experiment was carried out at 293 K.

### 2.3. UV–Visible Spectroscopy

UV–vis spectra were recorded using the Shimadzu UV-3600 spectrometer; the cell thickness was 1 cm, slit width 1 nm. Recording of the absorption spectra of the mixtures of moxifloxacin hydrocloride (moxi) and benzalkonium chloride (BCl) with pillar[5]arenes **3** at 1 × 10^−4^ M were carried out after mixing the solutions at 298 K. The 1 × 10^−4^ M solution of pillar[5]arene **3** (100, 120, 150, 200, 400, 600, 800 µL) in THF was added to 10 µL of the solution of guest (moxi) (1.2 ×10^−2^ M) in methanol and diluted to final volume of 3 mL with THF. The UV spectra of the solutions were then recorded. The stability constant of complexes were calculated as described below. Three independent experiments were carried out for each series. Student’s *t*-test was applied in statistical data processing. Experiment was carried out according to the literature method [22].

### 2.4. Dynamic Light Scattering (DLS)

The particle size and zeta potential was determined by the Zetasizer Nano ZS instrument at 20 °C. The instrument contains 4 mW He-Ne laser operating at a wave length of 633 nm and incorporated noninvasive backscatter optics (NIBS). The measurements were performed at the detection angle of 173°, and the software automatically determined the measurement position within the quartz cuvette. The 1 × 10^−4^−1 × 10^−6^ M THF solutions of **3**, the 1 × 10^−3^−1 × 10^−5^ M solutions of **3/3S** and **3/4S**, the 1 × 10^−4^ M solutions of antibiotics (moxi, BCl) (dissolved in methanol 1 × 10^−2^ M), and the complexes of macrocycle **3** or polymers **3/3S** or **3/4S** with antibiotics (moxi, BCl) were prepared. The concentration ratio of macrocycle **3** or polymers **3/3S** or **3/4S** and antibiotics in complexes was 1:10. The experiments were carried out for each solution in triplicate.

### 2.5. Transmission Electron Microscopy (TEM)

TEM analysis of samples was carried out using the JEOL JEM 100CX II transmission electron microscope. For sample preparation, 10 μL of the suspension 10^−5^ M were placed on the Formvar/carbon-coated 3 mm cuprum grid, which was then dried at room temperature. After complete drying, the grid was placed into the transmission electron microscope using special holder for microanalysis. Analysis was held at the accelerating voltage of 80 kV in SEM mode by Carl Zeiss Merlin microscope. Additionally, studies of the morphology of the samples were carried out using the Atomic force microscope Dimension FastScan (Bruker).

### 2.6. Gel Permeation Chromatography (GPC)

GPC studies were carried out on a GTsP chromatograph (Prague, Czech Republic) equipped with a refractometric detector and a 7.8 × 300 mm column. THF was used as the eluent. Phenogel 5 μm, pore size 10 Å (Phenomenex, CA, USA), was utilized as a sorbent. Polystyrene standards were used for calibration.

### 2.7. Simultaneous Thermogravimetry and Differential Scanning Calorimetry (TG–DSC)

TG–DSC was performed on a Netzsch Jupiter STA 449 C Jupiter analyzer in 40 μL platinum crucibles with a cap having a 0.5 mm hole at constant heating rates (10 and 4 deg/min; heating range 311–500 K) in dynamic argon atmosphere, flow rate 20 mL/min, atmospheric pressure; sample weight 10–20 mg. The results were processed using the NETZSCH Proteus software.

### 2.8. X-ray Diffraction Analysis

The dataset for single crystal **3** was collected on a Rigaku XtaLab Synergy S instrument with a HyPix detector and a PhotonJet microfocus X-ray tube using Cu Kα (1.54184 Å) radiation at low temperature. Images were indexed and integrated using the CrysAlisPro data reduction package. Data were corrected for systematic errors and absorption using the ABSPACK module: numerical absorption correction based on Gaussian integration over a multifaceted crystal model and empirical absorption correction based on spherical harmonics according to the point group symmetry using equivalent reflections. The GRAL module was used for analysis of systematic absences and space group determination. The structures were solved by direct methods using SHELXT [31] and refined by the full-matrix least-squares on F^2^ using SHELXL [32]. Non-hydrogen atoms were refined anisotropically. The hydrogen atoms were inserted at the calculated positions and refined as riding atoms. The positions of the hydrogen atoms of methyl groups were found using rotating group refinement with idealized tetrahedral angles. Disordered parts of the molecule are refined using constraints and restraints. The contribution of the disordered solvent was removed using the SQUEEZE option from PLATON operated the Olex2 interface. The figures were generated using Mercury 4.1 [33] program. Crystals were obtained by slow evaporation method.

Crystal data for C_55_H_70_O_10_S_10_ (*M* = 1211.71 g/mol): monoclinic, space group P2_1_/n (no. 14), *a* = 13.1087(6) Å, *b* = 12.0684(3) Å, *c* = 41.0482(14) Å, *β* = 92.459(4)°, *V* = 6487.9(4) Å^3^, *Z* = 4, *T* = 100.00(10) K, μ(Cu Kα) = 3.559 mm^−1^, *Dcalc* = 1.241 g/cm^3^, 46,894 reflections measured (4.31° ≤ 2θ ≤ 154.11°), 13,152 unique (*R*_int_ = 0.0519, *R*_sigma_ = 0.0522), which were used in all calculations. The final *R*_1_ was 0.1881 (I > 2σ(I)), and *wR*_2_ was 0.5275 (all data). CCDC refcode: 2163077.

### 2.9. Computational Method

Models of moxifloxacin and the thiolated pillar[5]arene were generated using Spartan ’20 [34]. The lowest energy structures were determined by molecular mechanics and geometry optimized by molecular mechanics using the Merck Molecular Force Field (MMFF). The 2:1 pillar[5]arene:moxifloxacin complex was created using these structures and geometry optimized using the MMFF followed by DFT (B3LYP/6-31G*) to give the final structure.

### 2.10. Biological Experiments

Films of free and moxi-loaded polymers **3/3S** and **3/4S** were formed in the wells of 8-well glass plates and dried for 72 h until a dry film formed on the adhesive surface of the glass.

Cultures of *Staphylococcus aureus* and *Klebsiella pneumoniae* were grown in L-broth to a density of 1.5 × 10^11^ cells/mL. Then, 400 μL were added to the wells of culture plates and chambers for microscopy. Cultivated at 37 °C for 48 h until the formation of stable biofilms.

Biofilms formed after 48 h of cultivation were washed with sodium phosphate buffer (pH = 7.2) from planktonic cells, dried under sterile conditions, and stained with 0.1% gentian violet solution for 20 min. Stained biofilms were washed three times with sodium phosphate buffer (pH = 7.2) and dried. Biofilm thickness was determined by washing with 96% ethanol the gentian violet dye from the biofilm matrix.

Light absorption measurements in eluate samples were measured at λ = 570 nm on a BIO-Rad xMark Microplate spectrophotometer.

The data were given in relative units. In the calculations, the light absorption of the eluate was taken as a unit in the variants without surface modification by films.

Most chemicals were purchased from Aldrich and used as received without additional purification. Organic solvents were purified in accordance with standard procedures.

### 2.11. Synthesis

Pillar[5]aren **1** were synthesized according to the literature procedure [35].

#### 2.11.1. Synthesis of 4,8,14,18,23,26,28,31,32,35-deca-[Acylthioethoxy]-pillar[5]arene (**2**)

Potassium thioacetate (0.74 g, 6.47 mmol) with anhydrous DMF (12 mL) were placed in a round-bottom flask equipped with a magnetic stirrer. The solution was stirred until complete dissolution. Then pillar[5]arene **1** 0.4 g (0.32 mmol) was added in one portion. Then the reaction mixture was heated for 56 h at 90 °C in an argon atmosphere. After the reaction, the mixture was poured into distilled water. The precipitated beige precipitate was filtered off on a Schott filter and washed with distilled water. The organic phase was separated and evaporated to dryness on a rotary evaporator.

Yield: 0.46 g (88%), mp. 100 °C. ^1^H NMR (CDCl_3_): 2.38 s (30H, –CH_3_), 3.30 t (20H, ^3^*J*_HH_ = 5.9 Hz, –CH_2_S–), 3.72 s (10H, –CH_2_–), 3.93–4.19 m (20H, –OCH_2_–), 6.79 s (10H, ArH). ^13^C NMR (CDCl_3_): 29.37; 29.57; 30.78; 67.55; 115.55; 128.53; 149.45; 195.34. IR (ν/CM^−1^) 2932 (–C_Ph_-H), 2868 (–CH_2_–, C_Ph_–O–CH_2_), 1684 (C=O), 1496 (–CH_2_–), 1465 (C_Ph_–C_Ph_), 1352 (–CH_3_), 1204 (C_Ph_–O–CH_2_), 1102 (C_Ph_–O–CH_2_), 1027 (C_Ph_–C_Ph_), 879 (–C_Ph_–H), 703 (C–S). MS (MALDI–TOF): calc. [M^ + ^] *m*/*z* = 1632.1, found [M+Na] ^+^ *m*/*z* = 1654.6. Found (%): C, 54.98; H, 5.46; N, 19.95. Calc. for C_75_H_90_O_20_S_10_. (%):C, 55.19; H, 5.56; O, 19.64; S, 19.64.

2932 (–C_Ph_–H), 2868 (–CH_2_–, C_Ph_–O–CH_2_), 1684 (C=O), 1496 (–CH_2_–), 1465 (C_Ph_-C_Ph_), 1352 (–CH_3_), 1204 (C_Ph_–O–CH_2_), 1102 (C_Ph_–O–CH_2_), 1027 (C_Ph_–C_Ph_), 879 (–C_Ph_–H), 703 (C–S).

#### 2.11.2. Synthesis of 4,8,14,18,23,26,28,31,32,35-deca-[2-Mercaptoethoxy]-pillar[5]arene (**3**)

In a round-bottom flask equipped with a magnetic stirrer, pillar[5]arene **2** 0.46 g (0.28 mmol) was dissolved in anhydrous acetonitrile (23 mL). Then hydrazine hydrate 0.98 mL (31.46 mmol) was added dropwise, and a white precipitate formed. The reaction was carried out for a week at room temperature in an argon atmosphere. Then, the reaction mixture was filtered and washed with acetonitrile. The precipitate was dissolved in chloroform and evaporated on a rotary evaporator in an argon atmosphere. The resulting white powder is the target product.

Yield: 0.25 g (75%), mp. 158 °C. ^1^H NMR (CDCl_3_): 1.65 t (10H, ^3^*J*_HH_ = 7.9 Hz, -SH), 2.78–2.83 m (20H, –CH_2_S–), 3.80 s (10H, –CH_2_–), 3.98 t (20H, ^3^*J*_HH_ = 5.3 Hz, –OCH_2_–), 6.78 s (10H, ArH). ^13^C NMR (CDCl_3_): 29.85; 29.90; 70.59; 115.80; 128.84; 149.85. IR (v, sm^−1^) 2931 (–C_Ph_–H), 2863 (–CH_2_–, C_Ph_–O–CH_2_), 2552 (–SH–), 1496 (–CH_2_–), 1463 (C_Ph_–C_Ph_), 1200 (C_Ph_–O–CH_2_), 1100 (C_Ph_–O–CH_2_); 1025 (C_Ph_–C_Ph_); 877 (–C_Ph_–H); 702 (C–S). MS (MALDI–TOF): calc. [M^+^] *m*/*z* = 1210.2, found [M + K + 2H]*^+^ m*/*z* = 1252.1, [M + Na + H]^+^ *m*/*z* = 1235.2. Found (%): C, 54.78; H, 5.98; S, 26.04, Calc. for C_55_H_70_O_10_S_10_. (%): C, 54.52; H, 5.82; O, 13.20; S, 26.46.

#### 2.11.3. General Procedure for the Synthesis of **3n**, **3/3S**, **3/4S**

A total of 0.09 g (0.074 mmol) of pillar[5] arene **3** was dissolved in 6 mL of THF; then, 0.35 mL of polythiol (trimethylolpropane tris(3-mercaptopropionate) or pentaerythritol tetrakis(3-mercaptopropionate)) (or without for **3n**) dissolved in 6 mL of THF and 1.2 mL of H_2_O_2_ (30%) dissolved in 6 mL THF was added. The reaction proceeded at room temperature for 40 h. Then the reaction mixture was poured into water and centrifuged. The resulting precipitate was dissolved in THF. Then, the precipitate was dried in vacuum under reduced pressure. The target product is a white, stretching mass.


*4,8,14,18,23,26,28,31,32,35-deca-[2-mercaptoethoxy]-pillar[5]arene (**3**)-based cross-linked supramolecular polymer (***3n***).*


Yield: 0.08 g (89%), mp. 317 °C. IR (v, sm^−1^) 2917 (–C_Ph_–H); 2859 (C_Ph_–O–CH_2_); 1496 (–CH_2_–); 1463 (C_Ph_–C_Ph_); 1194 (C_Ph_–O–CH_2_); 1058 (C_Ph_–O–CH_2_); 772 (C_Ph_–H); 698 (C–S).

*Tetrablock co-monomer, based on 4,8,14,18,23,26,28,31,32,35-deca-[2-mercaptoethoxy]-pillar[5]arene (**3**) and trimethylolpropane tris(3-mercaptopropionate), (***3/3S***)*.

Yield: 0.36 g (70%). MS (MALDI-TOF): calc. [M + 5K + 4Na + 1 × 3S-11H]^+^ *m*/*z* = 1886.7, [M + 5K + Na + 2 × 3S-12H]^+^ *m*/*z* = 2212.1, [2M + 10K + 2 × 3S-18H]*^+^ m*/*z* = 3589.6, [3M + Li + 3 × 3S-11H]^+^ *m*/*z* = 4825.8, found [M + 5K + 4Na + 1 × 3S-11H]^+^ *m*/*z* = 1887.4, [M + 5K + Na + 2 × 3S-12H]^+^ *m*/*z* = 2212.4, [2M + 10K + 2 × 3S-18H]^+^ *m*/*z* = 3590.2, [3M + Li + 3 × 3S-11H]^+^ *m*/*z* = 4825.8.


*Tetrablock co-monomer, based on 4,8,14,18,23,26,28,31,32,35-deca-[2-mercaptoethoxy]-pillar[5]arene (**3**) and trimethylolpropane tris(3-mercaptopropionate), (*
**3/4S**
*).*


Yield: 0.41 g (76%). MS (MALDI-TOF): calc. [2M + 3K + 3Na + 2 × 4S-14H]^+^ *m*/*z* = 3572, [2M + Na + 2 × 4S-1H]^+^ *m*/*z* = 3413.5, found [2M + 3K + 3Na + 2 × 4S-14H]^+^ *m*/*z* = 3572.0, [2M + Na + 2 × 4S-1H]^+^ *m*/*z* = 3413.4.

#### 2.11.4. General Procedure for the Synthesis of Cross-Linked Supramolecular Polymers **3/3Sn**, **3/4Sn**

Freshly prepared **3/3S** and **3/4S** tetrablock co-monomers (m = 0.3 g) were dissolved in 10 mL of THF, sprayed onto the surface of a glass substrate, and dried in the presence of atmospheric oxygen at ambient temperature for 30–40 min. Next, the formed supramolecular polymeric film **3/3Sn**, **3/4Sn** was used for further study.


*Cross-linked supramolecular polymer, based on 4,8,14,18,23,26,28,31,32,35-deca-[2-mercaptoethoxy]-pillar[5]arene (**3**) and trimethylolpropane tris(3-mercaptopropionate), (*
**3/3Sn**
*).*


Mp. 322 °C. IR (ν/CM^−1^): 3450 (–C=O); 2958 (–CH_2_–); 2941 (–C_Ph_–H); 2907 (C_Ph_–O–CH_2_); 2568 (–S–H); 1729 (–C=O); 1499 (–CH_2_–); 1471 (C_Ph_–C_Ph_); 1406 (–CH_2_–C=O); 1233 (–CH_2_–O–C=O); 1192 (C_Ph_–O–CH_2_); 1178 (C_Ph_–O–CH_2_); 1142 (–CH_2_–O–C=O); 1048 (C_Ph_–O–CH_2_); 671 (C–S); 583 (–S–S–).


*Cross-linked supramolecular polymer, based on 4,8,14,18,23,26,28,31,32,35-deca-[2-mercaptoethoxy]-pillar[5]arene (**3**) and pentaerythritol tetrakis(3-mercaptopropionate), (*
**3/4Sn**
*).*


Mp. 324 °C. IR (v, sm^−1^) 3451 (–C=O); 2960 (–CH_2_–); 2925 (–C_Ph_–H); 1726 (–C=O); 1497 (–CH_2_–); 1470 (C_Ph_–C_Ph_); 1407 (–CH_2_–C=O); 1230 (–CH_2_–O–C=O); 1209 (C_Ph_–O–CH_2_); 1173 (C_Ph_–O–CH_2_); 1127 (–CH_2_–O–C=O); 663 (–C–S–); 595 (–S–S–).

Detailed information of physical-chemical characterization is presented in Electronic Appendix A (ESI).

## 3. Results

### 3.1. Synthesis and Polymerization of Pillar[5]arene Containing Mercapto Groups

Self-healing is an attractive properties of materials, which is currently in demand [36]. The ability of a product to self-heal significantly increases its service life due to improved mechanical characteristics, surface renewal, and preservation of its integrity [37]. The design of such materials includes two main approaches to self-regeneration: the use of physical methods of cross-linking based on the mutual diffusion of individual parts and chemical methods of cross-linking using the formation of a reversible covalent bond [38]. In all of these approaches, macrocyclic compounds can be used. However, unlike self-healing through supramolecular interactions, chemical methods of self-healing provide higher mechanical strength and material stability [39].

Thus, to create self-healing materials based on pillar[5]arene, we chose to modify the macrocyclic platform with substituents able to form reversible covalent bonds. These interactions include the reversible formation of disulfide bonds, metal–ligand interactions, ionic interactions, and the formation of hydrogen bonds [40]. A common disadvantage of many such systems is the need for external influences, such as heating, UV irradiation, or pH changes through the addition of acid or alkali, which is necessary to initiate surface regeneration. The use of thiol/disulfide redox dynamic exchange reactions to form reversible disulfide bonds is the most accessible approach to date [40]. Although thiol/disulfide redox reactions can be accelerated in the presence of catalysts, they can also occur under ambient conditions (air temperature from 16 °C to 32 °C and relative humidity from 20% to 80%) with atmospheric oxygen as an external trigger [41].

To initiate thiol/disulfide cross-linking redox reactions in a polymeric material based on pillar[5]arene, the presence of thiol fragments in the structure of the macrocyclic platform is necessary. To this end, we developed an approach for introducing thiol fragments into the pillar[5]arene structure (Figure 1). Decabromoethoxy pillar[5]arene **1** was prepared according to the literature procedure [35] and reacted with potassium thioacetate in anhydrous DMF at 90 °C, whereupon macrocycle **2** was isolated by precipitation from water in 88% yield. Acetate fragments were cleaved with hydrazine hydrate in anhydrous acetonitrile [42] to give target macrocycle **3**, which was collected by filtration in 75% yield. Pillar[5]arene **3** was used without further purification in subsequent experiments (Figure 1, see Appendix A). The presence of free mercapto groups in macrocycle **3** was confirmed by one-dimensional ^1^H NMR spectroscopy, where they are observed as a triplet of 10 SH-protons at δ = 1.65 ppm (see Appendix A). It should be noted that the macrocycle **3** forms as a powder, which showed no oxidation over a month of storage under argon at room temperature. Macrocycle **3** was characterized using X-ray diffraction (Figure 1) from crystals grown from a CHCl_3_–CH_3_CN solvent mixture. The crystal habit of **3** is monoclinic, and the symmetry group is P 21/n.

Afterwards, we developed a procedure to prepare co-monomers and polymers using thiol/disulfide redox dynamic exchange reactions involving pillar[5]arene **3**. The polymerization proceeded under the action of 30% H_2_O_2_ in THF for 40 h at room temperature, and polymer **3n** was isolated in 88% yield (Figure 1) as a light-yellow powder (Figure 1). Polymer **3n** is practically insoluble in both polar and nonpolar solvents, and its decomposition onset temperature was 298 °C. Thus, it can be concluded that it is not suitable for the formation of self-healing films.

For the synthesis of self-healing films based on pillar[5]arene **3**, we chose to prepare cross-linked copolymers in the presence of low-molecular cross-linking agents: trimethylolpropane tris(3-mercaptopropionate) **3S** or pentaerythritol tetrakis(3-mercaptopropionate) **4S** (Figure 1). These commercially available compounds are used as gel formers [43] and polymer resin hardeners [44]. Thus, we carried out the selection of conditions for oxidative copolymerization to obtain copolymers in advance (see Appendix A).

Variation in the nature of the solvent (CH_3_CN, DMF, THF) and oxidants (I_2_, FeCl_3_) does not lead to the formation of polymer products **3** with **3S** or **4S**. Analysis of the ^1^H NMR spectra of the reaction mixtures showed the presence of the starting compounds and polymer **3n**. The use of H_2_O_2_ as an oxidizing agent and various ratios of reacting components led to the formation of comonomers **3/3S** and **3/4S**. The temperature regime of the syntheses varied in the range from 0–75 °C. However, the best results were achieved when the reaction was carried out at 25 °C for 24 h both for **3S** and **4S**. Thus, by reacting macrocycle **3** with 5-fold excesses of **3S** or **4S** in the presence of 30% H_2_O_2_ in THF for 24 h at room temperature, products **3/3S** and **3/4S** were isolated in 70% and 76% yields (see Appendix A).

The structure of the formed products **3/3S** and **3/4S** was studied by gel permeation chromatography (GPC) and MALDI mass spectrometry (see Appendix A). We chose GPC as a convenient method to determine the relative molecular weight of **3/3S** and **3/4S**. Thus, GPC analysis of fractions of samples **3/3S** and **3/4S** freshly prepared in THF showed average mass values up to 3000 Da, which corresponds to diblock- or tetrablock-linked (Figure 1) fragments of macrocycle **3** with polythiols **3S** or **4S** (see Appendix A). The formation of cross-linked tetrablock comonomers **3/3S** and **3/4S** can be explained by the polyfunctionality of the macrocyclic platform, the possible thermodynamic stability of the resulting tetrablock comonomers, and the low solubility of longer polymer units in THF [40].

The overall pattern of sequential fragmentation in the MALDI mass spectra of **3/3S** and **3/4S** tetrablock comonomers (see Appendix A) also agrees with the GPC data, as the mass spectra of **3/3S** and **3/4S** contain peaks of molecular ions in the range from 1713 Da to 5219 Da, which corresponds to a possible crosslinking from two to six fragments of **3S**, **4S,** and macrocycle **3** (see Appendix A).

The method of forming a film from a solution of co-monomers is a relatively simple process since the co-monomer is in a dissolved state. We used the method of spraying a solution of **3/3S** and **3/4S** in THF (1 × 10^–3^ M) [45] over the surface of a glass substrate. As a result, it was found that as the solvent evaporates, the solution passes into a gel-like state, followed by the formation of a film upon drying.

### 3.2. Interaction of Macrocycle ***3*** with the Antibiotic Moxifloxacin Hydrochloride

The films obtained on the basis of **3/3S** and **3/4S** contain fragments of pillar[5]arenes, which are capable of host–guest interactions with therapeutic drugs [26]. In this regard, we hypothesized that using an antimicrobial drug as a therapeutic agent would promote the formation of pillar[5]arene/drug complexes in the film structure to effectively suppress the development of bacteria. In addition, the presence of disulfide bonds (from **3/3S** and **3/4S**) in the film structure will contribute to the self-healing of the damaged surface under the action of atmospheric oxygen.

On account of the ability of macrocycle **3** to interact with antimicrobial drugs, benzalkonium chloride and moxifloxacin hydrochloride (moxi) were studied using UV–vis and NMR spectroscopy (see Appendix A). The choice of substrates was determined by their use in medical practice as effective preparations to treat bacterial infections [46]. The studies were carried out in a mixture of THF:CH_3_OH = 100:1. The choice of the solvent system was due to the good solubility of **3** in THF and its low solubility in solvents (water, alcohols), which dissolve binding substrates benzalkonium chloride and moxifloxacin hydrochloride.

It turned out that only when macrocycle **3** bound moxifloxacin hydrochloride were spectral changes at the absorption wavelength of moxifloxacin (λ = 340 nm) significant enough to establish quantitative binding characteristics (see Appendix A). The association constant was determined based on spectrophotometric titration data. Thus, the concentration of moxi (1 × 10^–5^ M) was constant, while the concentration of macrocycle **3** (0–2.67 × 10^–5^ M) varied (see Appendix A). Binding constants of **3**/moxi in THF:CH_3_OH = 100:1 were calculated by UV–vis spectroscopy from the analysis of binding isotherms and were established on the binding model **3**/moxi = 2:1 using Bindfit [47], a statistical model widely used in supramolecular chemistry to determine the characteristics of intermolecular interactions [48]. To confirm the proposed stoichiometry, the titration data were also processed using a binding model with a host–guest ratio of 1:1 and 1:2. However, in this case, the constants were determined with a much larger error (see Appendix A). The calculated logarithms of the association constants (logKa) for **3**/moxi were logK_11_ = 2.14 and logK_12_ = 6.20.

We chose 2D ^1^H-^1^H NOESY and 2D DOSY NMR (see Appendix A) spectroscopy to confirm the formation of the **3**/moxi complex and set its structure. An analysis of the experimental data obtained using ^1^H NMR spectroscopy did not make it possible to determine the nature of the interaction by changing the position of the host–guest chemical shifts. The chemical shifts of protons of moxi and **3** were broadened due to ongoing association processes (ESI). However, in the 2D ^1^H-^1^H NOESY NMR spectrum of associate **3**/moxi (2:1, C_moxi_ = 5 × 10^−3^ M) in CHCl_3/_CD_3_OD cross peaks between protons of aromatic fragments (**H^a^**) of macrocycle **3** and protons **H^10^**, **H^12^**, **H^13^** of the octahydro-1H-pyrrolo[3,4-b]pyridine fragment were observed (see Appendix A). Cross peaks are also observed between the protons of the cyclopropyl fragment of moxi (**H^4^** and **H^5^**) and the protons of the methylene bridges **H^b^**. The 2:1 inclusion complex was supported by calculations (DFT/BLY3P/6-31G*) as shown in Figure 2.

The formation of the **3**/moxi complex was additionally confirmed by 2D DOSY NMR spectroscopy. Diffusion coefficients of **3**, **3**/moxi, and moxi at 298 K (5 × 10^−3^ M) were determined. The 2D DOSY NMR spectrum of the **3**/moxi system shows the presence of signals of the complex lying on one straight line with one diffusion coefficient (D = 1.4 × 10^−10^ m^2^ s^−1^) (see Appendix A). This value of the diffusion coefficient **3**/moxi is much lower than the self-diffusion coefficients of macrocycle **3** (D = 3.5× 10^−10^ m^2^ s^−1^) and moxi (D = 4.7× 10^−10^ m^2^ s^−1^) under the same conditions. The results obtained definitely indicate the formation of an associate **3**/moxi. The formation of the **3**/moxi complex is in good agreement with the literature data on the binding of pillar[5]arenes with fluoroquinolone derivatives [49].

It is also important to evaluate the possibility of the interaction of moxi with **3/3S** and **3/4S** tetrablock co-monomers, which are soluble in THF and contain fragments of pillar[5]arene. Experiments on the interaction of moxi with **3/3S** and **3/4S** were carried out in THF:CH_3_OH (100:1). Highly sensitive fluorescence spectroscopy (Figure 3) was chosen as an effective method for detecting interactions between **3/3S**, **3/4S**, and the antibiotic moxi. The molecular weights of tetrablock co-monomers obtained by GPC (see Appendix A) to calculate the concentrations of **3/3S** and **3/4S**.

In the spectrum of moxi (5 μM), as **3/3S** (Figure 3a) and **3/4S** (Figure 3b) increases from 0.1 to 50 μM, the fluorescence also rises sequentially. The study was carried out at the emission wavelength of moxi (λ = 455 nm) in THF:CH_3_OH (100:1). Thus, the data of fluorescence spectroscopy confirm the interaction of **3/3S** and **3/4S** with the antibiotic.

This increase in the intensity emission can be related with the antibiotic’s inclusion into the cavity of pillar[5]arene, which is part of the tetrablock comonomer. The data obtained by 2D NOSY and DOSY NMR spectroscopy confirm this hypothesis.

For the **3/3S** and **3/4S**, the processes of self-association and aggregation in the presence of moxi were additionally studied by dynamic light scattering (DLS) in THF:CH_3_OH (100:1) (see Appendix A). It was shown that **3/3S** and **3/4S** do not form stable self-associates (see Appendix A) in the studied concentration range (1 × 10^−3^–1 × 10^−5^ M). However, adding a 10-fold excess of moxi to the tetrablock co-monomers **3/3S** or **3/4S** reduces the polydispersity index to 0.28–0.34. Stabilization of the system is observed when the average particle size of 460 nm is reached in the case of the moxi/**3/4S** (see Appendix A) system and 620 nm for moxi/**3/3S** (see Appendix A). Separately, macrocycle **3** and moxi did not form stable associates in THF:CH_3_OH (100:1) over the entire concentration range studied (1 × 10^−3–^1 × 10^−5^ M). When the method of spraying a solution of moxi/**3/3S** and moxi/**3/4S** in THF (1 × 10^−3^ M) onto a glass substrate was used, the formation of a drug-loaded films was observed.

### 3.3. Formation of Supramolecular Polymer Networks

Thus, pillar[5]arene **3** and tetrablock comonomers **3/3S** and **3/4S** based on it were able to interact with the antimicrobial drug moxi and form host–guest complexes. As shown above, **3/3S** and **3/4S** are able to form films loaded with moxi. In this regard, it can be assumed that moxi can be placed in the structure of the film based on pillar[5]arene and contribute to the suppression of the development of pathogenic microorganisms as part of the polymer coating.

To confirm this hypothesis, the films obtained after spraying a solution of **3/3S** or **3/4S** in THF (1 × 10^−3^ M) on a glass substrate were additionally investigated by a number of physical methods. The resulting cross-linked copolymers based on macrocycle **3** and thiols **3S** or **4S** were transparent films (Figure 1) soluble in THF. However, after evaporation of the solvent, the film becomes insoluble in THF over ~10–20 min, apparently due to the formation of additional disulfide bonds under the action of atmospheric oxygen. To confirm the formation of additional disulfide bonds and cross-linking **3/3S**, **3/4S** into **3/3Sn**, and **3/4Sn** polymer network compositions (Figure 1) during film formation, the IR spectra of **3/3Sn** and **3/4Sn** were studied and compared to **3** and **3n**. The IR spectra of **3/3Sn** and **3/4Sn** (Figure 4) show characteristic bands for the structure of thiols **3S**, **4S** (1732 cm^–1^), and macrocycle **3** (2960, 2925, 1470, 1209 cm^–1^). The absence of free S-H bonds vibrations at ν = 2750 cm^−1^ as well as the presence of SS bonds vibrations at ν = 640 cm^−1^ [50] in the fingerprint region, which did not appear in the initial macrocycle **3**, confirm the formation of additional disulfide bridges [40] under the action of atmospheric oxygen when THF solutions of **3/3S** and **3/4S** comonomers are dried.

Thermogravimetric analysis is widely used to assess the phase and thermal characteristics of polymeric materials [51], including self-regenerating ones [52] (see Appendix A). The DSC curve for macrocycle **3** includes three processes, one of which corresponds to the largest weight loss (34%), and three stages of decomposition on the TG curve. This exo-process (see Appendix A) is observed over the temperature range of 261–285 °C, which is consistent with the oxidative S–S crosslinking that can occur in a sample with a change in temperature [53]. The TG curves of the **3/3Sn** and **3/4Sn** samples (see Appendix A) contain four or five weight-loss steps. The first stage (up to 65 °C) corresponds to the removal of residual solvent. The second stage is different for each **3/3Sn** and **3/4Sn** sample and varies over the temperature range 137–211 °C. This stage is accompanied by a loss of up to 7% of the mass and is associated with the evaporation of water included in the structure of the film. In contrast to **3** alone, the DSC curve in the temperature range of 250–300 °C of the **3/3Sn** and **3/4Sn** (see Appendix A) samples does not show an exo-process. This process corresponds to the main stage of weight loss. However, in a higher temperature range of 280–340 °C, in samples **3/3Sn** and **3/4Sn**, an endo-process is observed corresponding to the first stage of destruction (25–34% weight loss) and of the melting of substances **3/3Sn** and **3/4Sn**. The fourth step on the TG curve for the **3/3Sn** and **3/4Sn** samples corresponds to the weight loss of 13–17% and on the DSC curve in the temperature range of 347–374 °C corresponding of oxidative S–S cross-linking.

Thus, analysis of the TG-DSC (see Appendix A) data allows us to conclude that the **3/3Sn** and **3/4Sn** polymer structures are more thermally stable than macrocycle **3**, which is characterized by thermally sensitive S–S crosslinking processes. The melting temperature of free macrocycle **3** is 30–50 °C lower than that of **3/3Sn** and **3/4Sn**, which confirms the improved thermal characteristics of the obtained materials.

### 3.4. Interaction of ***3/3S***, ***3/4S*** and Supramolecular Copolymers ***3/3Sn***, ***3/4Sn*** with the Antibiotic Moxifloxacin Hydrochloride

Since **3/3Sn** and **3/4Sn** polymer films were formed from **3/3S** and **3/4S** tetrablock co-monomers, capable of interacting with moxi, it was necessary to investigate the ability to bind moxi in the structure of **3/3Sn** and **3/4Sn**. For this purpose, moxi/**3/3Sn**, moxi/**3/4Sn** polymer films were formed by spraying a solution of moxi/**3/3S** and moxi/**3/4S** (THF:CH_3_OH = 100:1, 1 × 10^−3^ M) on to the glass substrate for 30 min (Figure 3f,g) and the fluorescence spectra of the resulting films recorded (Figure 3c,d). It is known that ionized forms of antibiotics of the fluoroquinolone series dissolve well in water, therefore, under real conditions, when the environmental humidity changes, the antibiotic can be removed from the polymer surface. In order to simulate this situation, the surface of moxi/**3/3Sn** and moxi/**3/4Sn** was repeatedly washed with distilled water. The number of washes varied from 1 to 20 times. According to fluorescence spectroscopy data, the intensity of moxi emission in the samples decreased during the first washing for moxi/**3/3Sn** by 67% (Figure 3c) and by 45% for moxi/**3/4Sn** (Figure 3d). Further washing did not lead to significant changes in the moxi fluorescence intensity in the **3/3Sn**, **3/4Sn** films.

Since the moxi/**3/4Sn** systems turned out to be the most monodisperse and resistant to washing, their morphology was studied using electron and atomic force microscopies (see Appendix A). According to scanning electron microscopy (SEM) data, the **3/4Sn** film is an irregular network polymer consisting of intertwining filaments with a thickness of 63 nm (Figure 5a). A similar morphology is confirmed by 3D images of the atomic force microscope (AFM) (Figure 5b). Also, according to transmission electron microscopy (TEM) (Figure 5c, see Appendix A), the formation of dendritic structures on the surface of interlacing threads is observed. The TEM images of moxi/**3/4Sn** show the formation of dendritic structures typical of polymer morphology with included spherical moxi particles on the surface (Figure 5d). The size of spherical particles moxi was 100 nm, and the thickness of the dendritic fragments corresponded to the thickness of the filaments and was 67 nm (Figure 5d, see Appendix A).

Thus, in the course of the studies, the formation of **3/3Sn**, **3/4Sn** films capable of binding antibacterial drugs of the fluoroquinolone series was demonstrated. The resulting **3/3Sn**, **3/4Sn** structures are built on the principle of the formation of dynamic -S-S- covalent bonds, which, due to thiol/disulfide redox dynamic exchange reactions, lead to self-healing of damaged surface areas. The mechanism of self-healing process is proposed to be through the formation of free sulfur radicals [54], which again form disulfide bonds under the action of external triggers (Figure 6a).

### 3.5. Study of the Process of Self-Regeneration of Films ***3/3Sn***, ***3/4Sn***

Electron paramagnetic resonance (EPR) was used to determine the presence of sulfur radicals in **3/3S**, **3/4S**, **3n**, **3/3Sn**, **3/4Sn** structures, since free radicals have a nonzero electronic (spin) magnetic moment, characterized by a quantum number of S = 1/2 [55]. Figure 6b shows the steady-state absorption spectrum for the **3n** powder sample at room temperature. Comparing the spectroscopic g-factor value with the literature data [54,55], it is obvious that sulfur radicals are present in the **3n** (see Appendix A) powder sample, which confirms the hypothesis put forward.

No EPR signals in liquid **3/4S** samples with different THF concentrations were observed, which may indicate the instability of sulfur radicals in these solutions. The heating of **3n** powder also did not lead to the formation of new sulfur radicals. The shape of EPR spectrum has a weak (but visible) asymmetry, which indicates the anisotropy of the g-factor arising from the interaction of the electron shell of the radical with the surrounding electric field gradients of neighboring ions. The phase coherence time (T_M_) of 578 ns is a rather short value for transverse relaxation compared to other stable radicals [56]. The asymmetry of the spectrum (anisotropy) and the short value of T_M_ (due to nuclear spin diffusion) are associated with the localization of the sulfur radical in the polymer structure.

Additionally, samples of **3/4Sn** and **3/3Sn** in the form of film structures (see Appendix A) were studied. The EPR spectra show signals from sulfur radicals superimposed on a broader and structureless line. The formation of this line (underlayer signal) is possibly associated with a change in the structure or local environment of the sulfur radical, which leads to a strong (dipole-dipole) inhomogeneous broadening.

Since radiation exposure leads to the formation of stable free radicals, a sample of the **3/4Sn** film was undergone by X-ray irradiation (Figure 6c). To improve the signal-to-noise ratio (20 times), the experiments were carried out at 15 K. After irradiation, the spectrum is a sum of signals of different origins, which is possibly due to the presence of a hyperfine interaction of the paramagnetic center with the nuclei of hydrogen ^1^H or sulfur ^33^S or the formation of other types of sulfur radical. Figure 6c also shows the dynamics of the EPR spectra that decrease in intensity with time. This indicates the lower stability of these paramagnetic centers, in contrast to the radicals in the powder sample, the spectrum of which does not change with time.

The **3/3Sn** sample was also irradiated with an X-ray source for 1 h, but this procedure did not lead to the formation of additional EPR signals.

Modern microscopy methods are convenient and effective tools for dynamic study of the self-healing process [57]. Therefore, the process of self-healing of the formed films based on the **3/4Sn** system was qualitatively assessed using optical microscopy to monitor the healing of the cut surface under the action of atmospheric oxygen. Initially, the 3D surface of a **3/4Sn** film was created by SEM at low pressure. An uneven thickness distribution in the film upon drying in air was found (Figure 7a). Micrometer-sized surface scratches were then made on the surface using a micrometer blade (Figure 7b). The film was stored in the ambient atmosphere, and the damaged area was monitored using an optical microscope (Figure 7b, see Appendix A). The healing response was clearly observed within 2 h at room temperature. The cut healed from the ends where the cut surfaces were closest to each other. In this case, the cut surface, according to SEM data, was an intergrowth of dendritic structures directed perpendicular to the cut wall (Figure 7c). This not only demonstrates the ability of the material to heal, but also opens up the possibility of creating a material with the function of self-regeneration under the action of biological substrates or in the body’s environment.

### 3.6. Antibacterial Properties of Self-Regenerating Films ***3/3Sn***, ***3/4Sn***

Despite the ability of **3/4Sn** to self-heal and interact with moxi, the antibacterial drug may not be available for entry into the bacterial matrix. As a result, the moxi/**3/4Sn** system will not inhibit the formation of microbial biofilms.

It is worth noting that, according to the literature data [58], pillar[5]arenenes containing thioether fragments do not have pronounced cytotoxicity. This makes these compounds attractive for use in biomedical materials.

To test the ability of a moxi/**3/3Sn** and moxi/**3/4Sn** films to suppress the development of pathogenic microorganisms, we evaluated the formation of bacterial biofilms on the adhesive surfaces of slide chambers treated with moxi/**3/3Sn** and moxi/**3/4Sn**. Pathogenic microorganisms of the Gram-negative morphotype of the *Enterobacteriaceae* family [59] and Gram-positive bacteria *Staphylococcaceae* have serious impacts on human health. Methicillin-resistant strains of *Staphylococcus aureus* attract special attention in the clinic [60]. In this study, a clinical isolate of *Klebsiella pneumonia* belonging to the *Enterobacteriaceae* family and *Staphylococcus aureus* ATCC^®^ 29213™ were selected as model pathogens capable of forming biofilms.

Modification of the surface of adhesive glasses with **3/4Sn**, **3/3Sn** and moxi/**3/4Sn**, moxi/**3/3Sn** led to a change in the thickness of microorganism biofilms (Figure 8b). On the surface coated with **3/4Sn** and **3/3Sn** films, in the case of *S. aureus* biofilms and biofilm of *K. pneumonia*, a slight increase in the total biomass of biofilm was observed compared to the untreated variant (Figure 8a). A significant scatter of data in this processing option can be associated with a different area of the modified surface. Addition of moxi into **3/4Sn** and **3/3Sn** films reduced the total biomass of biofilm of both *S. aureus* and *K. pneumoniae*.

Thus, moxi/**3/4Sn** and moxi/**3/3Sn** films were formed by sputtering solutions of tetrablock co-monomers in THF (1 × 10^−3^ M) in a chamber with an adhesive glass bottom. The resulting moxi/**3/4Sn** and moxi/**3/3Sn** systems were washed five times with distilled water to remove excess unbound moxi. It was shown that moxi/**3/4Sn** reduced the capacity of biofilms formed by *S. aureus* and *K. pneumoniae*, by 80% and 48%, respectively (Figure 8a). In the variant with the moxi/**3/3Sn** film (1 × 10^−3^ M), its application reduced the capacity of the biofilm formed by *S. aureus* and *K. pneumoniae*, by 77% and 43%, respectively (Figure 8a).

Additionally, the inhibition of pathogenic biofilm formation by *S. aureus* and *K. pneumoniae* in the presence of moxi was studied (Figure 8a). An analysis of experimental data showed that individual moxi suppresses the development of pathogenic biofilms more effectively. However, moxifloxacin hydrochloride is highly soluble in water, which makes moxi not applicable under changing environmental conditions. When humidity changes, moxi will be washed off the treated surface. Doping moxi into the composition of the polymer film makes it possible to keep it on the surface and create a concentration gradient near the biofilm. It should be noted that the efficiency of moxi in the composition of the polymer film remains as high as without it (Figure 8).

Drug release methods are as important as the encapsulation process. As a mechanism for releasing moxi from the moxi/**3/4Sn** complex, a guest exchange mechanisms can be assumed [61]. So, a molecule with a high affinity for the macrocyclic pillar[5]arene should be chosen as a new guest. As such molecules, some amino acids can be selected like arginine (Arg) and lysine (Lys). These amino acids are part of the human body proteins—elastin and collagen. Arginine (Arg) and lysine (Lys) have a large affinity for the carboxylated pillar[5]arene cavity [62]. The process of pathogenic biofilm formation on the surface of moxi/**3/4Sn** will gradually release the encapsulated moxi, thereby ensuring continuous maintenance of the concentration of the active form of moxifloxacin hydrochloride in the biofilm matrix and in close proximity to microbial cells.

## 4. Conclusions

A novel decasubstituted pillar[5]arene containing free mercapto groups, **3**, was synthesized and its structure determined by powder X-ray diffraction. Using UV–vis spectroscopy, the ability of pillar[5]arene **3** to interact with the antimicrobial drug moxifloxacin was shown. The association constant and stoichiometry of the **3**/moxi complex were calculated by UV–vis spectroscopy from the analysis of binding isotherms for model **3**/moxi = 2:1 (logK_11_ = 2.14 and logK_12_ = 6.20). The structure of the resulting complex was confirmed by 2D ^1^H-^1^H NOESY NMR spectroscopy. THF soluble tetrablock co-monomers **3/3S** and **3/4S** were isolated by thiol/disulfide redox reactions of **3** with trimethylolpropane tris(3-mercaptopropionate) **3S** or pentaerythritol tetrakis(3-mercaptopropionate) **4S**, the structure of which was studied by GPC and MALDI mass spectrometry. Spraying of **3/3S** and **3/4S** solutions in THF (1 × 10^−3^ M) on the surface of a glass substrate led to the formation of **3/3Sn** and **3/4Sn** polymer films. The formation of films occurs due to the formation of additional -S-S- bonds between tetrablock co-monomers **3/3S** and **3/4S**. They were characterized by TG-DSC analysis, FTIR spectroscopy, and their morphology studied by electron microscopy. Optical spectroscopy and EPR showed that the resulting **3/4Sn** film had the ability to self-heal under atmospheric oxygen. It was found that the **3/3Sn** and **3/4Sn** systems do not affect the formation of biofilms formed by *S. aureus* and *K. pneumoniae*. However, the introduction of the drug moxi into the composition of **3/3Sn** and **3/4Sn** films resulted in a noticeable inhibition of the formation of biofilms of these pathogenic microorganisms. The ability to retain an antimicrobial drug in **3/3Sn** and **3/4Sn** films after washing with water was shown by fluorescence spectroscopy. These results open up wide opportunities to develop new antibacterial polymeric materials with self-healing abilities that are resistant to external conditions.

## Figures and Tables

**Figure 1 nanomaterials-12-01604-f001:**
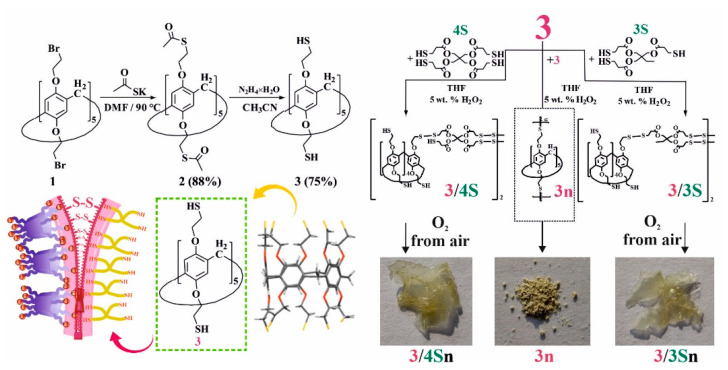
Synthesis of macrocycles **2** and **3**; X-ray lateral view of macrocycle **3** and sketch showing copolymerization of macrocycle **3** into **3n** and **3** with trimethylolpropane-tris(3-mercaptopropionate) **3S** and pentaerythritol-tetrakis(3-mercaptopropionate) **4S** in THF in the presence of 5 wt. % H_2_O_2_ and atmospheric oxygen.

**Figure 2 nanomaterials-12-01604-f002:**
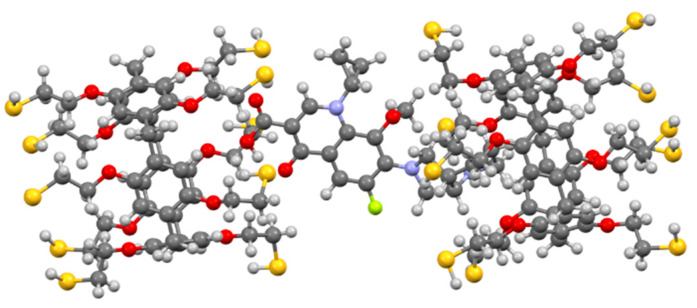
Geometry optimized structure of the **3**/moxi complex.

**Figure 3 nanomaterials-12-01604-f003:**
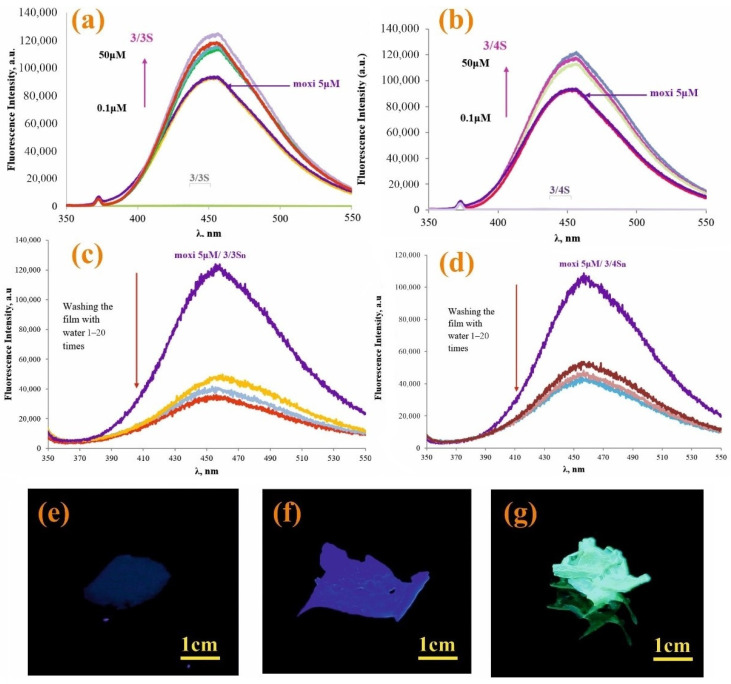
Fluorescence spectra of moxi (5 × 10^−6^ M) with various concentrations of: (**a**) **3/3S** (0–50 μM) and (**b**) **3/4S** (0–50 μM); (**c**) fluorescence spectra **3/3Sn**/moxi and (**d**) **3/4Sn**/moxi of the film before and after washing with distilled water; photographs of samples under UV irradiation at λ = 365 nm (**e**) **3n**; (**f**) **3/4Sn**; and (**g**) **3/4Sn** + moxi after washing with water (five times).

**Figure 4 nanomaterials-12-01604-f004:**
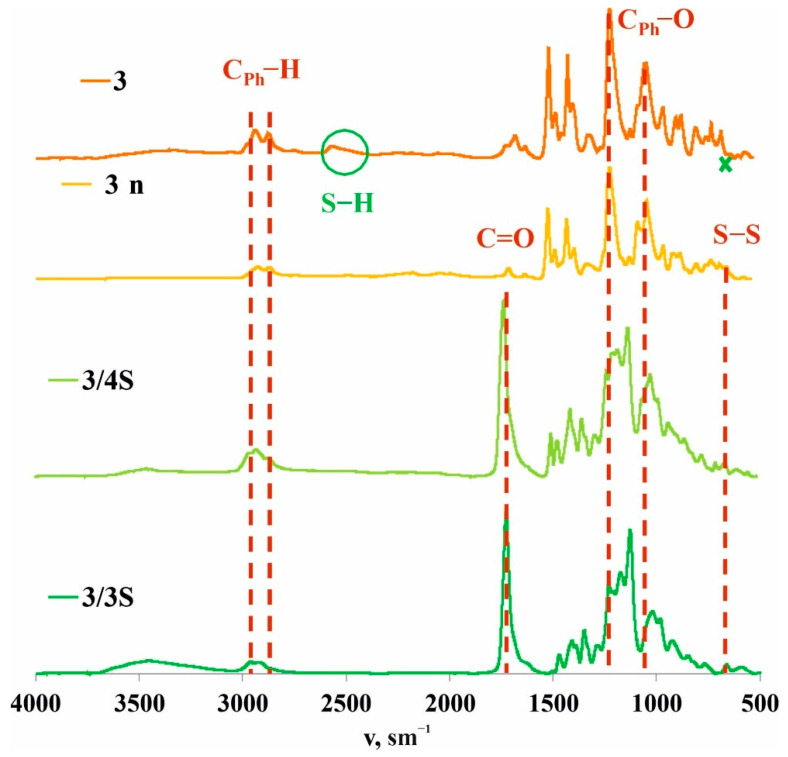
Attenuated total internal reflectance IR spectra of **3** and **3n** powders and **3/3Sn** and **3/4Sn** films.

**Figure 5 nanomaterials-12-01604-f005:**
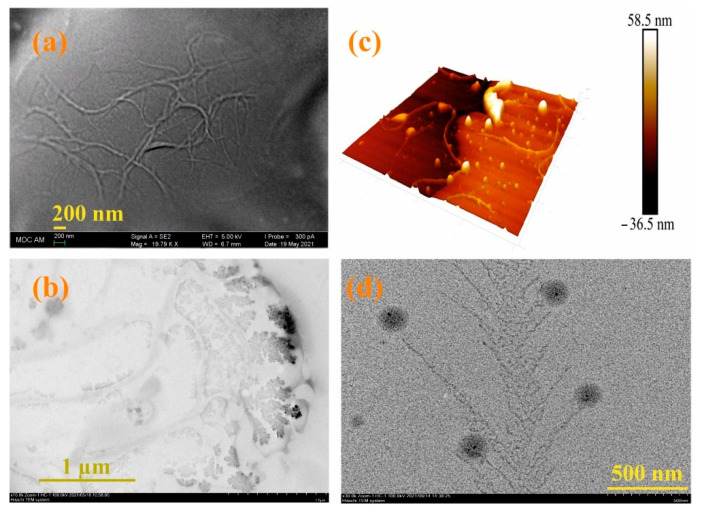
(**a**) SEM images of **3/4Sn** (1 × 10^−5^ M) after the solvent (THF:CH_3_OH (100:1)) evaporation; (**b**) AFM images of a **3/4Sn** film (1 × 10^−5^ M) after the solvent (THF:CH_3_OH (100:1)) evaporation; (**c**) TEM images of **3/4Sn** film (1 × 10^−5^ M) after the solvent (THF:CH_3_OH (100:1)) evaporation; (**d**) TEM images of system **3/4S** (1 × 10^−5^ M)/**moxi** (1 ×10^−4^ M) after the solvent (THF:CH_3_OH (100:1)) evaporation.

**Figure 6 nanomaterials-12-01604-f006:**
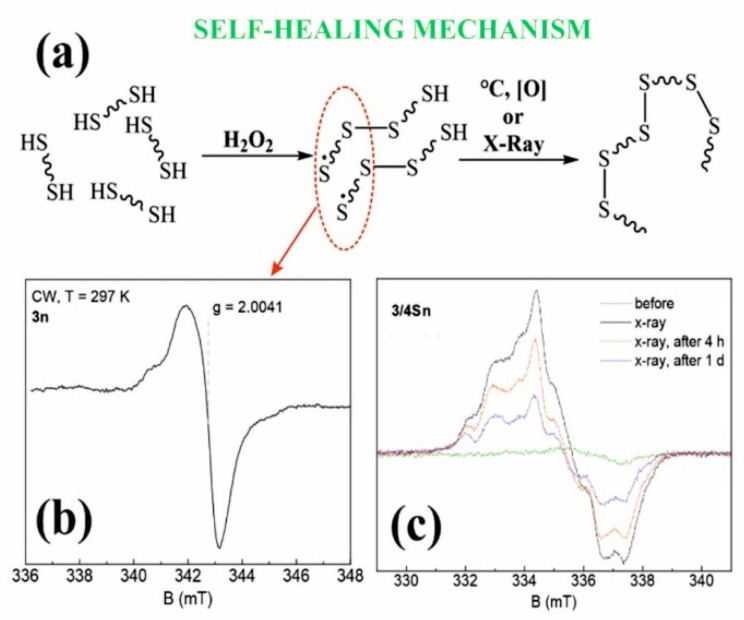
(**a**) Proposed mechanism of self-regeneration; (**b**) Stationary electron paramagnetic resonance spectrum of sample 3n (powder) at room temperature in the X-band continuous wave mode (9.6 GHz); (**c**) EPR spectra of **3/4Sn** films at T = 15 K before and after X-ray irradiation.

**Figure 7 nanomaterials-12-01604-f007:**
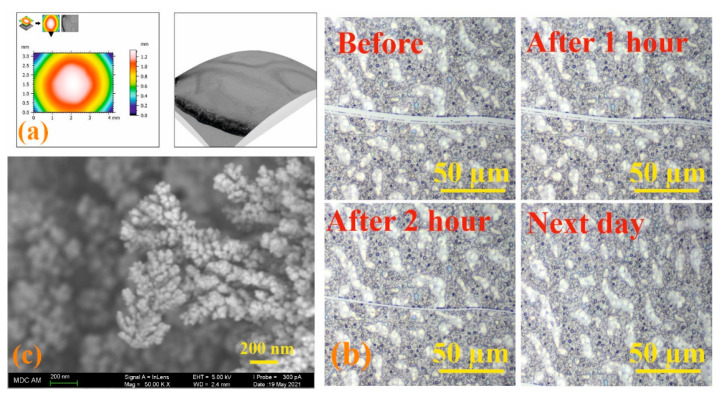
(**a**) SEM images of **3/4Sn** at low pressure and topographic map of **3/4Sn** film; (**b**) optical microscope image of a **3/4Sn** film with surface disturbance over time (0–24 h); (**c**) SEM image of a section of a **3/4Sn** film.

**Figure 8 nanomaterials-12-01604-f008:**
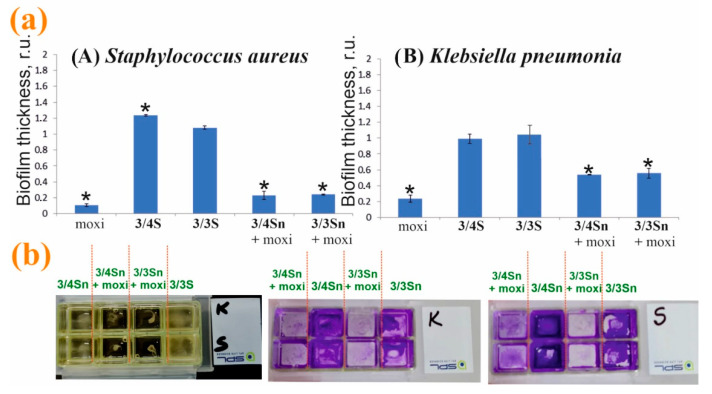
(**a**) Effect of pretreatment of the adhesive glass surface with **3/3Sn**, **3/4Sn** films and free moxi, as well as their composites with moxi on the ability to form *S. aureus* and *K. pneumoniae* biofilms. The power of the biofilm of microorganisms in the variant without pretreatment was taken as a unit. (**b**) Photographs of slide chambers with cultures of bacteria in the presence of **3/4Sn** and **3/3Sn** and moxi/**3/4Sn**, moxi/**3/3Sn**. *—*p* ≤ 0.05 when compared with the variant without pre-treatment of the surface.

## Data Availability

Not applicable.

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
