# Peer review of "Self-Healing Thiolated Pillar[5]arene Films Containing Moxifloxacin Suppress the Development of Bacterial Biofilms"

_nanomaterials, 2022, doi:10.3390/nano12091604_

Round 1

Reviewer 1 Report

The manuscript is based on well-established experimental research and is clearly written and interesting. Only the minor changes, labeled directly in the text, are nedeed.

Author Response

Many thanks to the respected reviewer for the review of our manuscript. However, we did not receive the manuscript with marked corrections.

Reviewer 2 Report

1. Figure 3e-g suggest adding a scale bar.
2. The differential analysis of figure 8 was not observed on the figure.
3. There is only one group in figure 8a that differs greatly between the values, have you analyzed the possible reasons?
4. It is suggested to adjust the typography of figure 8b to more clearly observe the differences between groups.

Author Response

  1. « Figure 3e-g suggest adding a scale bar.»

Answer:

We have included the proposed changes in the manuscript.

The figure 3e-g was changed in the manuscript:

  1. « The differential analysis of figure 8 was not observed on the figure. »

Answer:

Differential analysis has been added to the figure 8.

  1. « There is only one group in figure 8a that differs greatly between the values, have you analyzed the possible reasons?»

Answer:

Indeed, there is a group in Figure 8a that greatly differs between the values. This is due to the fact that during the initial analysis of the results, all values of the experiment, including extremes, were included. When drop-down points (extremes) are removed, the differences between the values become less noticeable. Figure 8a has been corrected.

  1. « It is suggested to adjust the typography of figure 8b to more clearly observe the differences between groups.»

Answer:

We have included the proposed changes in the manuscript.

The figure 8 was changed in the manuscript:

Reviewer 3 Report

Authors prepared the self-healing films containing fragments of pillar[5]arene obtained using thiol/disulfide redox cross-linking for anti-bacterial coating. Authors characterized the films using thermogravimetric analysis and differential scanning calorimetry, FTIR spectroscopy and electron microscopy. Authors also performed the UVvis, 2D 1H- 1H NOESY and DOSY NMR analysis to investigate the complexes of the pillar[5]arene with the antimicrobial drug moxifloxacin. Authors showed also the films containing moxifloxacin(moxi) effectively reduced Staphylococcus aureus and Klebsiella pneumoniae biofilms formation on adhesive surfaces.

Although authors performed several instrumental analysis for the development of moxi containing films, ability to suppress the biofilm formation of microorganisms need to be more clearly investigated. How can authors explain the big difference (77 vs 43%) of reduction capacity of the biofilm formation by S. aureus and K. pneumonia? Authors need to perform more experiments for more bacterial pathogens. What is the reduction mechanism to suppress the biofilm formation? How can you explain the release of moxi from the films in aqueous system like bacterial cultures?  How about the cytotoxicity of synthetic films? What would be critical advantage of self-healing film in this research? Complexation analysis of pillar[5]arene with moxi were performed in the organic solvent rather than aqueous system. How could you explain the moxi release mechanism for the suppression of pathogen bacterial biofilm? How about the suppressing effects between the direct treatment of moxi to bacterial biofilms compared to the moxi-containing films treatment to the bacterial biofilms?

Author Response

  1. « How can authors explain the big difference (77 vs 43%) of reduction capacity of the biofilm formation by S. aureus and K. pneumonia?»

Answer:

Indeed, as a result of the experiment, it was shown that the moxi/3/4Sn film reduces the capacity of biofilms formed by S. aureus and K. pneumoniae by 80% and 48%, respectively. In the case of the moxi/3/3Sn film, its use reduced the capacity of the biofilm formed by S. aureus and K. pneumoniae by 77 and 43%, respectively.

Since our experiment on studying the changes in the biofilms thickness of microorganisms S. aureus and K. Pneumoniae in the presence of 3/4Sn and 3/3Sn did not show a decrease in the capacity of biofilms. But on the contrary, there was a slight increase in the total biomass of the biofilm compared to the variant without treatment. Thus it can be said that the moxifloxacin hydrochloride in the moxi/3/4Sn and moxi/3/3Sn films has an inhibitory effect. Earlier in the literature, it was shown that moxifloxacin showed markedly better activity against S. aureus and S. epidermidis compared to E. coli and K. pneumoniae. However, moxifloxacin is highly effective against both E. Coli, K. Pneumoniae and S. Aureus, S. Epidermidis [Lemmen, S. W.; Häfner, H.; Klik, S.; Lütticken, R.; Zolldann, D. Comparison of the bactericidal activity of moxifloxacin and levofloxacin against Staphylococcus aureus, Staphylococcus epidermidis, Escherichia coli and Klebsiella pneumoniae. Chemotherapy. 2003, 49, 33-35; Callegan, M. C.; Ramirez, R.; Kane, S. T.; Cochran, D. C.; Jensen, H. Antibacterial activity of the fourth-generation fluoroquinolones gatifloxacin and moxifloxacin against ocular pathogens. Adv. Ther. 2003, 20, 246-252]. Our results are in good agreement with the literature data.

  1. « Authors need to perform more experiments for more bacterial pathogens. What is the reduction mechanism to suppress the biofilm formation? »

Answer:

This work represents the first such study in which host-guest complexes are part of self-healing polymer films based on the formation of reversible covalent bonds. First of all, it was an exploratory study to evaluate the effectiveness of the use of antibacterial drugs as part of a self-healing film. In connection with this, we have chosen the most typical opportunistic microorganisms: S. aureus and K. Pneumoniae. Of course, our further work will be directed to the study of a larger number of bacterial pathogens. As for the mechanism of restoration of the suppression of biofilm formation, then, as mentioned above, the mechanism is associated with the specific action of the antibiotic, moxifloxacin hydrochloride.

  1. « How can you explain the release of moxi from the films in aqueous system like bacterial cultures?»

Answer:

Drug release methods (moxifloxacin hydrochloride) are important as much as the encapsulation process. A detailed study of moxi release methods may be the subject of a new publication. In this manuscript, we report the first example of the use of self-healing films based on pillar[5]arene with bound moxifloxacin hydrochloride as pathogen control systems. However, the reviewer has asked a right question. A guest exchange mechanisms (M. D. Pluth and K. N.  Raymond, Chem. Soc. Rev., 2007, 36, 161) can be assumed for releasing moxi from the complex moxi/3/4Sn. Thus, a molecule with a high affinity to the macrocyclic pillar[5]arene should be chosen as a new guest. Some amino acids, such as arginine (Arg) and lysine (Lys), which are a part of the proteins elastin and collagen, can be selected as such molecules. Arg and Lys show a large affinity toward carboxylated pillar[5]arene cavity. Similar system was described by I. Bitter et al. (M. Bojtár, A. Paudics, D. Hessz, M. Kubinyi and I. Bitter, RSC adv., 2016, 6, 86269). For example, aminonaphthalimide derivatives with various anchor groups in the 4-position formed a 1:1 complex with the pillar[5]arene. Association constants (Kass) for the resulting complexes ranged from 103 to 106. As a result of the complexation process, a strong quenching of the fluorescence of the synthesized aminonaphthalimide derivatives was observed. However, in the presence of arginine and lysine, the aminonaphthalimide derivative / pillar[5]arene complex decomposed and an arginine or lysine molecule was embedded in the macrocycle cavity, while the fluorescence of the synthesized aminonaphthalimide derivatives was restored. It is worth noting that the supramolecular system that we collected also contains pillar[5]arene, which also can potentially have an affinity for aliphatic amino acids with pronounced base properties.

The following text was added to the manuscript:

“Drug release methods are as important as the encapsulation process. As a mechanism for releasing moxi from the moxi/3/4Sn complex, a guest exchange mechanisms can be assumed (M. D. Pluth and K. N.  Raymond, Chem. Soc. Rev., 2007, 36, 161). So, a molecule with a high affinity for the macrocyclic pillar[5]arene should be chosen as a new guest. As such molecules, some amino acids can be selected like arginine (Arg) and lysine (Lys). These amino acids are part of the human body proteins - elastin and collagen. Arginine (Arg) and lysine (Lys) have a large affinity for the carboxylated pillar[5]arene cavity (M. Bojtár, A. Paudics, D. Hessz, M. Kubinyi and I. Bitter, RSC adv., 2016, 6, 86269). The process of pathogenic biofilm formation on the surface of moxi/3/4Sn will gradually release the encapsulated moxi, thereby ensuring continuous maintenance of the concentration of the active form of moxifloxacin hydrochloride in the biofilm matrix and in close proximity to microbial cells.”

  1. « How about the cytotoxicity of synthetic films?»

Answer:

Of course, the characterization of cytotoxicity is important for any drug. Previously, a number of pillar[5]arenes containing thioether fragments were synthesized in our research group [Shurpik, D. N.; Aleksandrova, Y. I.; Mostovaya, O. A.; Nazmutdinova, V. A.; Zelenikhin, P. V.; Subakaeva, E.V.; Mukhametzyanov, T. A.; Cragg, P. J.; Stoikov, I. I. Water-soluble pillar[5]arene sulfo-derivatives self-assemble into biocompatible nanosystems to stabilize therapeutic proteins. Bioorg. Chem. 2021, 117, 105415.] and it was shown that all these macrocycles do not have pronounced cytotoxicity against A549 model cells in the entire range of concentrations studied (0.5–50 µg/ml). It should also be noted that we do not consider moxi/3/4Sn systems as individual drugs. These systems can be used as biomedical materials in the fight against pathogenic microorganisms, without direct contact with the human body.

The following text was added to the manuscript:

“It is worth noting that, according to the literature data [Shurpik, D. N.; Aleksandrova, Y. I.; Mostovaya, O. A.; Nazmutdinova, V. A.; Zelenikhin, P. V.; Subakaeva, E. V.; Mukhametzyanov, T. A.; Cragg, P. J.; Stoikov, I. I. Water-soluble pillar[5]arene sulfo-derivatives self-assemble into biocompatible nanosystems to stabilize therapeutic proteins. Bioorg. Chem. 2021, 117, 105415.], pillar[5]arnenes containing thioether fragments do not have pronounced cytotoxicity. This makes these compounds attractive for use in biomedical materials.”

  1. « What would be critical advantage of self-healing film in this research?»

Answer:

The main advantage of our material is the unique combination of self-healing and pathogen control functions. Such a symbiosis of properties in one polymer coating will make it possible to create a material that can withstand changing external conditions without losing its antibacterial properties. In turn, the presence of a macrocyclic system in the composition of the film makes it more versatile. The interaction of the film and the antibacterial drug does not occur covalently, but due to the formation of a host-guest complex. This fact makes it possible to change the structure of the antibacterial drug depending on certain conditions, selecting a new antibacterial drug that is complementary to the macrocyclic cavity.

  1. « Complexation analysis of pillar[5]arene with moxi were performed in the organic solvent rather than aqueous system. How could you explain the moxi release mechanism for the suppression of pathogen bacterial biofilm?»

Answer:

The matrix of pathogenic biofilms is highly hydrated, according to some estimates, 97% consists of bound water. The main components of the matrix are polysaccharides, proteins and extracellular DNA. The composition of the matrix can vary greatly depending on the environmental conditions and the type of microorganisms. It is worth noting that moxifloxacin hydrochloride is water-soluble, unlike the 3/4Sn film components, which allows it to be inside the matrix.

The mechanism of the release of moxifloxacin hydrochloride from the moxi/3/4Sn complex can be assumed to be the mechanism of guest substitution. Thus, a molecule with a high affinity for the macrocyclic cavity of the pillar[5]arene should act as a new guest. Some amino acids, such as arginine (Arg) and lysine (Lys), which are part of the biofilm matrix proteins, can be chosen as such molecules.

  1. « How about the suppressing effects between the direct treatment of moxi to bacterial biofilms compared to the moxi-containing films treatment to the bacterial biofilms?»

Answer:

A study of the inhibition of pathogenic biofilm formation in the presence of moxi was inserted in the Figure 8. An analysis of experimental data showed that individual moxi suppresses the development of pathogenic biofilms more effectively. However, moxifloxacin hydrochloride is highly soluble in water, which makes moxi not applicable under changing environmental conditions. When humidity changes, moxi will be washed off the treated surface. Doping moxi into the composition of the polymer film makes it possible to keep it on the surface and create a concentration gradient near the biofilm. Near the surface of the polymer film, the concentration of moxi is maximum. Moxi action is localized directly near the biofilm matrix. Outside the biofilm matrix, moxi will be held in the film structure by intermolecular forces. It should be noted that the efficiency of moxi in the composition of the polymer film remains as high as that of free moxi.

The figure 3e-g was changed in the manuscript:

The following text was added to the manuscript:

“Additionally, the inhibition of pathogenic biofilm formation by S. aureus and K. pneumoniae in the presence of moxi was studied (Fig. 8). An analysis of experimental data showed that individual moxi suppresses the development of pathogenic biofilms more effectively. However, moxifloxacin hydrochloride is highly soluble in water, which makes moxi not applicable under changing environmental conditions. When humidity changes, moxi will be washed off the treated surface. Doping moxi into the composition of the polymer film makes it possible to keep it on the surface and create a concentration gradient near the biofilm. It should be noted that the efficiency of moxi in the composition of the polymer film remains as high as without it (Fig. 8). »

Round 2

Reviewer 3 Report

Now the revised manuscript can be accepted.